# Exploiting Virus Infection to Protect Plants from Abiotic Stresses: Tomato Protection by a Begomovirus

**DOI:** 10.3390/plants11212944

**Published:** 2022-11-01

**Authors:** Rena Gorovits, Moshe Shteinberg, Ghandi Anfoka, Henryk Czosnek

**Affiliations:** 1Institute of Plant Sciences and Genetics in Agriculture, Robert H. Smith Faculty of Agriculture, Food and Environment, The Hebrew University of Jerusalem, Rehovot 7610001, Israel; 2Faculty of Agricultural Technology, Al Balqa’ University, Al-Salt 10117, Jordan

**Keywords:** virus, tomato, environmental stresses, virus-plant interaction

## Abstract

Tomato cultivation is threatened by environmental stresses (e.g., heat, drought) and by viral infection (mainly viruses belonging to the tomato yellow leaf curl virus family—TYLCVs). Unlike many RNA viruses, TYLCV infection does not induce a hypersensitive response and cell death in tomato plants. To ensure a successful infection, TYLCV preserves a suitable cellular environment where it can reproduce. Infected plants experience a mild stress, undergo adaptation and become partially “ready” to exposure to other environmental stresses. Plant wilting and cessation of growth caused by heat and drought is suppressed by TYLCV infection, mainly by down-regulating the heat shock transcription factors, HSFA1, HSFA2, HSFB1 and consequently, the expression of HSF-regulated stress genes. In particular, TYLCV captures HSFA2 by inducing protein complexes and aggregates, thus attenuating an acute stress response, which otherwise causes plant death. Viral infection mitigates the increase in stress-induced metabolites, such as carbohydrates and amino acids, and leads to their reallocation from shoots to roots. Under high temperatures and water deficit, TYLCV induces plant cellular homeostasis, promoting host survival. Thus, this virus-plant interaction is beneficial for both partners.

## 1. Introduction

Environmental stresses affect crop production worldwide. Drought and heat are the most serious abiotic stresses especially in countries with hot climate. Moreover, they increase insect pressure (reviewed in [1]), creating additional threats due to the viruses carried by insect vectors. For a long time, viruses were considered exclusively as pathogens that cause damage to plants, using host resources and compounds for their own reproduction. However, in recent years, new information described the beneficial roles of some viruses for their plant hosts. Viruses were defined as mutual interactors with plant cellular mechanisms, resulting in plant protection against some abiotic stresses [2]. One of the pioneering studies in this direction described the ability of RNA viruses such as brome mosaic virus (BMV), tobacco mosaic virus (TMV), and tobacco rattle virus (TRV), to increase host plant tolerance to drought, while cucumber mosaic virus (CMV) induced plant resistance not only to drought, but also to cold [3]. The virus-infected tissues were characterized by increased levels of osmoprotectants and antioxidants. A further advancement in the understanding of drought tolerance was achieved in *Nicotiana*
*benthamiana* and *Arabidopsis thaliana* infected by potato virus X (PVX) and plum pox virus (PPV) [4]. Even though a detailed description of PVX or PPV-induced improvement of plant survival under drought was not presented, increased levels of salicylic acid (SA), but not abscisic acid (ABA), were found in the infected plants. These results were unexpected, because ABA is the key hormone regulating plant responses to drought [5], while SA is induced in response to infections by pathogens [6,7]. However, the establishment of drought tolerance in an ABA-independent manner was also shown in transgenic Arabidopsis expressing the tomato yellow leaf curl C4 gene [8] or the turnip mosaic virus (TuMV) 6K2 gene [9]. TRV was shown to promote plant tolerance to low temperature [10], and PVX to environmental oxidation [11]. 

Begomoviruses are small circular ssDNA (cssDNA) viruses. Tomato yellow leaf curl virus (TYLCV) is a typical Begomovirus, which infects tomato plants (*Solanum lycopersicum*). The TYLCV virion encapsidates a single cssDNA molecule (monopartite) of about 2800 nucleotides. Until very recently, it was thought that the monopartite TYLCV genome encodes six genes: the viral strand with two genes, V1 (coat protein CP) and V2, and the complementary strand with four genes, C1 to C4 (reviewed in [12]). However, recent bioinformatic studies [13] were instrumental in the discovery of six additional TYLCV open reading frames coding for proteins of less than 80 amino acids, some displaying specific subcellular localizations. For example, V3 expressed during viral infection, localizes in the Golgi apparatus, functions as an RNA silencing suppressor, and traffics along microfilaments to plasmodesmata to promote virus cell-to-cell movement [14]. Another TYLCV protein, coined C5, is a pathogenicity determinant and RNA silencing suppressor [15]. Another recent study [16] showed that by screening translation initiation sites of the bipartite tomato yellow leaf curl Thailand virus, potential viral proteins coded by the A and B genomes were discovered and found to be important for the translation of different protein isoforms localized in various cellular compartments.

TYLCV is spread between plants by its insect vector in a circulative persistent manner, the whitefly *Bemisia tabaci*, which feeds on more than 600 plant species worldwide (reviewed in [17]). In commercial tomatoes, yield losses may reach 100% [18]. The threat of TYLCV pathogenicity has been reduced by the development of tomato lines and cultivars tolerant to TYLCV, first in Israel [19] and later in many other countries [20]. Upon infection, these tolerant tomatoes (coined here R-TYLCV) do not change drastically their morphological characteristics, show normal growth and yield even under massive inoculation by viruliferous whiteflies. After prolonged infection, R-TYLCV tomatoes contain virus amounts comparable to those present in TYLCV-susceptible tomatoes (S-TYLCV), while conserving they agronomic properties [21]. 

TYLCV does not induce a hypersensitive response and cell death upon whitefly-mediated infection of S-TYLCV tomato plants, until diseased tomatoes become senescent [22]. On the contrary, the capacity of TYLCV to enhance the survival of infected tomatoes (S and R) exposed to different stresses has been gradually uncovered. It is summarized in the current review. 

## 2. Cell Death, Induced by Inactivation of HSP90, Is Suppressed in TYLCV-Infected Plants 

Cell death can often be activated as a defense response against biotic and abiotic stresses [23,24]. Silencing of the genes encoding the key cellular chaperone HSP90 and HSP90 co-chaperone, SGT1, caused necrosis in the tomato stem [25]. The inactivation of the Hsp90 machinery resulted in massive accumulation of ubiquitinated proteins. A decrease in necrotic lesions was conspicuous in TYLCV-infected plants where the Hsp90/Sgt1 genes have been silenced. Host protein damage decreased as well. 

The central role of HSP90 in protein damage was shown first in yeast [26], then in plants [27]. HSP90 develops complexes with 26S proteasome, activating the degradation of ubiquitinated proteins. Inactivation of the HSP90 machinery resulted in the dissociation of the 26S proteasome and in the loss of its peptidase activity. Likely, TYLCV mitigated the inactivation of 26S proteasome in Hsp90/Sgt1-silenced tomatoes, restoring the proper utilization of ubiquitinated proteins in cellular protein homeostasis, and leading to the suppression of cell death in virus-infected tomatoes. While uninfected tomatoes with silenced Hsp90/Sgt1 genes died after two weeks of growth, the infected tomatoes appeared healthy during prolonged growth [25] (Figure 1).

## 3. Interplay of Tomato Stress Response Machinery and TYLCV Infection Increased Plant Survival during Growth at High Temperatures

Numerous stresses cause protein damage and induce cellular protective mechanisms in order to maintain protein homeostasis [28]. These mechanisms are conserved in all eukaryotic organisms and are based on the activation of chaperones/HSPs functions. The regulation of HSPs is carried out by heat shock transcription factors (HSFs). 

The redundancy of genes involved in stress-response is a witness of the importance of such mechanisms for the survival of the organism. Genome-wide expression studies (in vivo and in silico) have shown that heat shock factors (including heat shock proteins and their transcription factors) are encoded by large gene families (about 21 in tomato), which play a role in heat stress responses [29,30]. Transcriptome sequencing indicated that subjecting tomato cv Moneymaker to short periods (hours) of heat stress was accompanied by an increase in HSP expression, e.g., HSP20 and HSP70 [31]. However, in the latter study, the redundancies of the HSP genes were not considered. It was neither in earlier HSP-HSFs interaction studies [32]. 

The number of plant HSFs genes is several-fold that of animals, suggesting that HSFs have additional functions in plants [33]. Indeed, the overexpression of HSFA2 in Arabidopsis results not only in an enhanced resistance to heat, but also to light [26]. In tomatoes, heat stress induces the expression of two key transcription factors, HSFA2 and HSFB1, in complement to the constitutively expressed HSFA1. HSFA2 is a stable protein, which reaches high levels under prolonged plant exposure to heat and even during plant recovery from a period of high temperature [34]. In the case of TYLCV-dependent regulation of HSPs patterns, the redundancy of HSPs is not a critical issue. TYLCV prevents dramatic changes in *Hsps* expression in plant cells, stabilizing the amount of HSPs. In our earlier studies, HSP stabilization was shown for several selected HSPs (HSP60, 70, 90), especially in TYLCV-infected R-tomatoes [21,22]. 

The expression of *HsfA2* and *HsfB1* were used as markers of tomato plant’s response to heat stress in the absence and presence of TYLCV. Uninfected tomatoes reacted to heat shock by an intensive induction of *HsfA2* and *HsfB1* transcription; moreover, the expression of HSFA2-dependent genes (*Hsp90*, *Hsp17*, *Apx1*, *Apx2*) were similarly induced by high temperatures. TYLCV-infected tomatoes exposed to heat showed an alleviated induction of *HsfA2*, *HsfB1*, *Hsp90*, *Hsp17*, *Apx1*, and *Apx2* [35]. Hence, virus infection suppressed the acute response of plant cells to heat stress, preventing cell death and allowing adaptation of the plant to high temperatures all along its development. 

In order to understand the need for such suppression, an analysis of soluble (active) and aggregated (inactive) HSP90 and HSP70 was carried out. It is a well-known phenomenon that increased temperatures are accompanied by unfolding and by nonspecific protein aggregation. Heat stress caused a re-localization of HSP90 and HSP70 from a soluble to an insoluble state, decreasing the activity of this key HSPs in tomato tissues [28]. Moreover, the aggregation was exacerbated in the presence of TYLCV. The appearance of virus-induced large protein aggregates, harboring viral proteins, mainly CP, and DNA, is a feature of a successful virus invasion in S-TYLCV tomatoes [36]. Heat-treated tomato tissues containing increased protein aggregation. Recruited cellular HSPs, mainly HSP70 and HSP90, were not completely rescued, even after prolonged recovery periods (Figure 2). We suggest that the virus-dependent down-regulation of heat stress response in tomatoes prevented the development of massive protein aggregation and allowed the release of free soluble active chaperones, leading to a restoration of cellular homeostasis. 

One of the possible mechanisms explaining the observed heat stress response down-regulation is related to the ability of viral proteins to interact with the plant HSFA2. We have previously described, for the first time, the interaction between HSFA2 and TYLCV CP in tomato [35]. The His-tagged TYLCV major viral proteins CP, V2, C1, C2, C3, and C4, were overexpressed in *E. coli* and bound to His-Bind Resin in Ni-column. Total protein extracts from heat shock-treated (1 h) infected tomatoes (14 dpi) were passed through Ni-columns, each bound to a different His-tagged viral protein. Following elution, the bound proteins were immuno-detected with anti-HSFA2 antibodies. Complexes between HSFA2 and all the six TYLCV proteins were detected. *E. coli* extracts were used as controls. By comparison, HSP70 interacted with only three viral proteins, including CP. To identify HSFA2-TYLCV complexes in the tomato leaf cellular compartments, proteins from separated cytoplasmic and nuclear fractions were passed through the Ni resin columns bound to the viral proteins. Pull down of plant HSFA2-CP complexes were found in nuclear, but not in cytoplasmic, protein extracts. HSFA2-CP binding was not found in *E. coli* protein extracts. These results indicated that TYLCV CP was able to bind cellular HSFA2, restricting the capacity of free transcription factor to promote heat shock gene transcription in plant nuclei. 

The exposure of eukaryotic cells to elevated temperatures leads to the translocation of inactive cytoplasmic HSFA2 into its active nuclear form, which binds promoters of heat stress dependent genes ([37], and references therein). TYLCV proteins, especially CP, maintained HSFA2 in the cytoplasm, preventing transcriptional activation of heat stress response genes [35]. 

Hsp101 is another downstream protein regulated by HsfA class members. HSP101/ClpB chaperone is induced by heat and other stresses in different plants [reviewed in [38]. In a previous study of tomato response to heat shock [35], we used anti-HSP101 antibodies to demonstrate the induction of HSP101 after heat shock; the activation of HSP101 was similar to that of HSFA2 and HSP90 and was dependent on TYLCV amounts contained in tomato tissues. In the current review, the role of HSP70 and HSP90 is discussed not only in response to heat stress, but mainly in the development of protein aggregates. We have not studied the involvement of HSP101 aggregates in tomatoes infected by TYLCV. 

## 4. TYLCV Infection of Tomatoes Enhances Tolerance to Drought

The first descriptions of a virus-dependent increase in plant tolerance to drought concerned RNA viruses such as BMV, CMV, TMV, and TRV [8]. PVX and PPV were also described to be beneficial for *N. benthamiana* and Arabidopsis grown under drought conditions [9]. The ability of TYLCV, particularly of its C4 protein, to promote enhanced tolerance to drought was reported in tomato and in *N. benthamiana*; however, a mechanism explaining this phenomenon was not proposed [13]. Indeed, TYLCV-infected tomatoes survived after 25–30 days of growth after water withhold, compared to approximately two weeks for uninfected tomatoes [39]. The observed resilience of virus-infected plants to drought was accompanied by a reallocation of key metabolites, from shoots to roots (Figure 3). By decreasing the metabolic activity of shoots, tomato reduced nutrient uptake. In contrast, in roots, which played a determining role in drought response, the uptake of metabolites was enhanced. The redistribution of metabolites is a major requirement for buffering the effects of drought in many plants. Drought causes a decrease in essential free sugars and amino acids in shoots, concomitant to an increase in these metabolites in roots [40]. The reallocation of carbohydrates and amino acids from shoots to roots suggests a role of roots in protecting infected tomatoes against drought [39].

In response to drought, similarly to the response to heat, the presence of TYLCV in plants caused a down-regulation of stress response proteins, including HSP90 and HSP70 [39]. Moreover, the expression of three key HSFs (*HsfA1*, *HsfA2*, *HsfB1*) was less induced by drought in infected than in uninfected plants (Figure 3). HSFs have multiple functions, beyond the activation of HSPs. In particular, HSFs are able to control various stress-induced processes, such as 26S proteasome degradation, apoptosis, autophagy, cell growth arrest and many others [42]. Tomato HsfA1a was shown to be an important regulator of drought tolerance through the induction of autophagy and activation of ATG genes (*ATG10* and *ATG18f*) [43]. In drought-treated tomatoes, the induction of HSFA1a was followed by increased levels of *ATG10* and *ATG18f* transcripts, especially in roots [39]. Geminivirus proteins are degraded by autophagy [44,45]. Indeed, a decline in TYLCV DNA and CP amounts accompanied tomatoes grown under conditions of water withholding, therefore, TYLCV caused a suppression of drought-activated autophagy. Comparing uninfected and TYLCV-infected tomatoes grown under conditions of water withholding indicated that virus infection caused a reduced rate of transpiration and a stabilization of plant water balance. This resulted in less water use from soil and promoted plant survival [39,41]. Attenuated transpiration was observed in different tomato cultivars and was ultimately dependent on the presence of the virus.

## 5. TYLCV Infection Mitigates the Activation of Stress Markers Associated with Carbamazepine (CBZ) Treatment

The world is experiencing an increased demand for clean water [46]. In many countries with hot and dry climate, agriculture crops face a water deficit, which could be alleviated by increasing the efficiency of water usage and by using treated wastewater for irrigation [47]. However, even treated wastewater still contains various toxic products, including pharmaceuticals, only partially removed during purification, which accumulate in soils [48]. CBZ, a drug used to treat human neurological disorders, is one of the abundant pollutants remaining in treated wastewater [49]. Plants irrigated with treated wastewater absorb CBZ by the roots, which subsequently spreads to the leaves by xylem and phloem transport [50]. The accumulation of CBZ causes typical stress response in different tomato tissues by activating stress markers such as HSPs, two isoforms of glutamate decarboxylase (GAD1 and 2) belonging to the GABA shunt enzymes, and osmo-protective metabolites [51]. Even though CBZ induced relatively weak stress responses in tomatoes, TYLCV was able to mitigate them. CBZ-dependent increase in soluble sugars levels in leaves and roots, the enhancement of stress-related amino acids and enzymes patterns in roots were diminished in virus-infected tomatoes in comparison to uninfected plants. The TYLCV mediated suppression of the mild CBZ-caused stress may be due to an autophagy activation, observed in CBZ-treated plants, which led to the degradation of viral proteins. 

## 6. Use TYLCV Infection to Induce Tomato Protection against Environmental Stresses 

The new concept, described in the current review, is based on the two apparently contradictory events: on the one hand, acute environmental stresses cause plant death; on the other hand, viruses inhibit cell death to create a proper environment for their successful replication and propagation. Moreover, TYLCV infection itself could cause a total loss of tomato crops [52]. To avoid pathogenic viral effects, we developed several TYLCV-resistant lines. Such a line, coined R-GF967, was able to sustain extreme heat [35]. R-GF967 seedlings were inoculated with TYLCV 5–7 days before transfer to greenhouses where heat reached 45–50 °C. TYLCV promoted tomato growth for much longer time compared to uninfected R-GF967 tomatoes. Virus pre-inoculation of the R-GF967, promoted extreme tolerance to drought [41]. Uninfected tomatoes (S- and R-TYLCV) ceased to grow after two weeks of complete water withholding, while these plants pre-infected with TYLCV continued to grow (Figure 4A). In the infected R-TYLCV plants, not only protein and osmolyte homeostasis, but also water balance reached steady levels. Moreover, recovery to regular watering resulted in fruit yield comparable to that of plants grown under standard irrigation (Figure 4B).

## 7. Conclusions

All modern agricultural plants are the result of selection and improvement of wild relatives of the cultivated species. While the wild relatives have acquired tolerance to the major abiotic and biotic stresses, the adaption to harsh environments caused them to be inedible and sometimes even poisonous. During the Neolithic agricultural revolution (about 10,000 years ago) the selection for better, tastier, edible plants began. The selection for yield did not start until the advent of modern breeding in the last 300 years, when the selection for more visual-appealing fruits and higher yields to feed increasing populations started after the black death and other epidemics in Europe, Asia and the Americas. Since bad quality and low yield were associated with genes providing resistance to abiotic and biotic stresses, the selection for fruit quality and high yield led to the loss of the resistance genes. As a result, today, the major agricultural plants (e.g., rice, maize, wheat, tomato) are very susceptible to biotic (viruses, fungi, bacteria) and abiotic (heat, drought, salt) stresses and require the intervention of man to survive by spraying pesticides, applying fertilizers, and by irrigation. 

Some solutions for these issues are as follows: -Breeding for resistance: finding the resistance genes from the wild relatives and putting them back in the cultivated plants by crosses and selection.-Genetic engineering: by influencing the metabolic pathways to increase resilience.-Overexpress in transgenic plants those viral proteins that have been shown to be involved in tolerance to abiotic stresses, such as 2b in CMV, C4 in TYLCV, or P25 in potato virus X [53].-Finding new ways to increase tolerance. The described approach of using DNA virus, such as TYLCV, can increase tomato resilience to several abiotic stresses.

One of the major effects caused by TYLCV infection of tomato is a decrease in the activation of stress response proteins and metabolites to avoid an acute deleterious plant response, which may impair virus replication. Instead of severe, sometimes lethal response for most plant cells, TYLCV promotes the development of a protective homeostasis response in plants exposed to prolonged environmental stresses, favoring the survival of the plant. The down-regulation of stress proteins and metabolites coincides with the stabilization of their patterns, particularly in R-TYLCV tomatoes [39,41]. These stable patterns are maintained not only in shoot, but also in roots of virus-infected tomatoes. Additional, more specific targets of virus-dependent suppression of tomato stress response may exist. For example, drought- and CBZ-induced autophagy was shown to be mitigated in the infected tomatoes, preventing degradation of all the TYLCV proteins otherwise prone to autophagy [35,51].

Whether fruit production of pre-inoculation of R-TYLCV tomato seedlings in field and greenhouses located in hot and dry countries provides a solution, even partial, to global warming, remains to be analyzed on a large scale. This routine, adapted to tomato genotype, soil and climate, may help reduce drastically the amount of water used for irrigation by artificially alternating long periods of drought with short periods of irrigation (recovery).

Pre-inoculation may present some disadvantages for tomatoes grown in the open field because of the danger of outbreaks from a resistant plant to sensitive genotypes (or other species that could be infected by TYLCV such as pepper, due to the efficient vector *B. tabaci* [54]). To avoid such a possibility, it might be possible to pre-inoculate (by agroinfection) seedlings with a TYLCV symptomless mutant lacking 20 amino acids near the N-terminus of the CP, and therefore not transmissible by whiteflies [55], before planting in the field. Of course, the stress-related effects of the mutants (and that of Agrobacterium) will have to be compared with those of the wild type virus. 

Pre-inoculation with a virus wild type may cause the appearance of mutants in the host plant. Viruses can adapt to a new host by modifying its genome to favor, for example, improved replication. It can also modify itself under various environmental stresses. Ecological selection of new virus variants has been recently discussed; they usually are not the result of the response of the host to infection but rather of changes affecting the spread of the virus insect vector or changes in vector populations following drastic environmental changes [56]. The mutation rate of TYLCVs in its tomato host has been reported to be in the order of 10^−4^ substitutions per site during the 45 days of the experiment. Most relevant to our work, the rate of mutation in TYLCV-susceptible and resistant genotypes was not significantly different [57].

Discovering the interactions between biotic and abiotic stresses, viruses, cultivated plants and the environment are of importance for ensuring our food security. Although progress has been made to understand these interactions, much is still unknown and should be investigated in the future.

## Figures and Tables

**Figure 1 plants-11-02944-f001:**
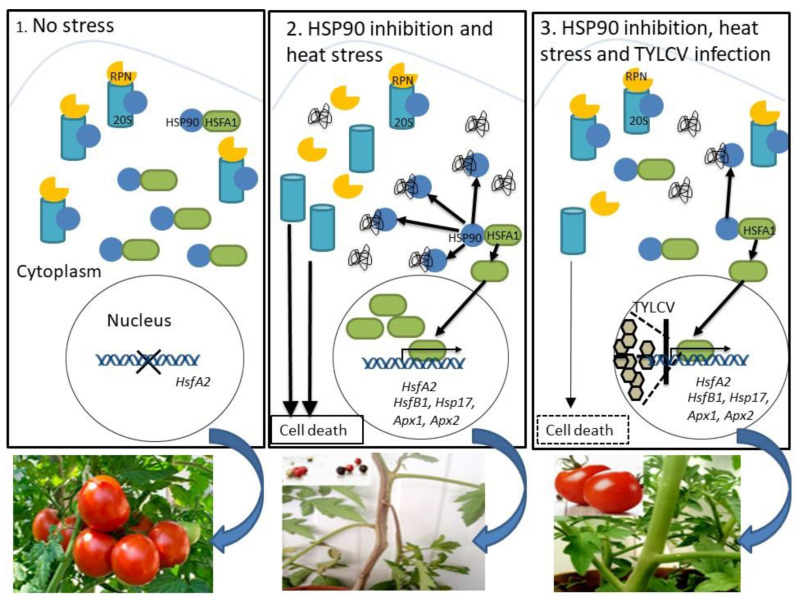
Summary of the key processes of stress response regulation by HSP90 in tomato plants, and their down-regulation by TYLCV infection. Functional loss of the cellular chaperone HSP90 causes the dissociation of the 26S proteasome and a significant decrease in its peptidase activity, consequently, to an increase in the level of ubiquitinated proteins and signs of cell death. The inhibition of the 26S proteasome stimulated the expression of heat-inducible genes, including transcription factor HSFA2. In leaves of TYLCV-infected tomato, the levels of HSFA2 were lower than in leaves of uninfected plants. The amounts of HSFA2 greatly increased upon heat stress in uninfected tissues, and much less in TYLCV-infected leaves. The inhibition of HSP90 activity caused an additional increase in HSFA2 expression. Subsequent TYLCV infection reduced HSFA2 levels as well as expression levels of HsfB1, Hsp17, Apx1, and Apx2. TYLCV infection suppresses HSP90-dependent 26S proteasome inactivation, cell death and HSFA2 signal transduction pathways, resulting in normal tomato growth and fruit yielding [25].

**Figure 2 plants-11-02944-f002:**
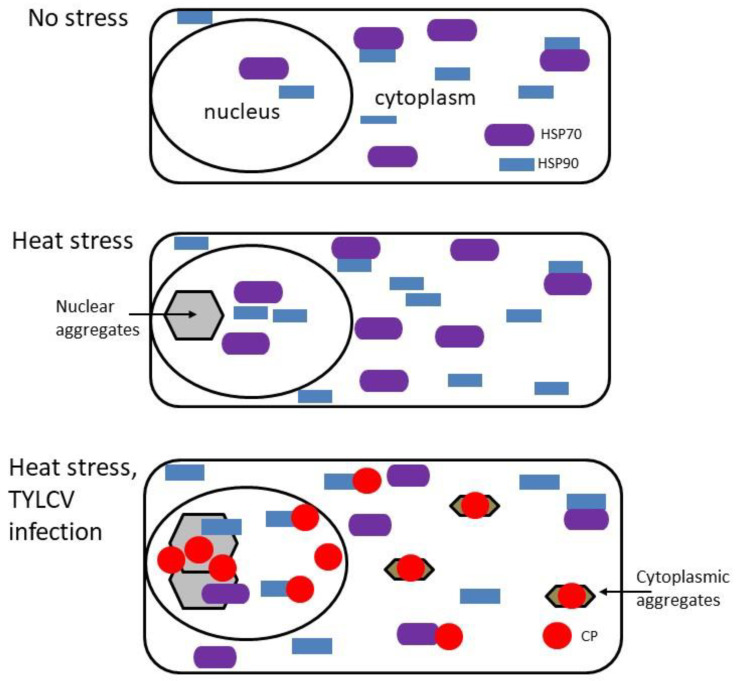
The diagram summarizes the association of TYLCV with tomato chaperones in aggregates. Unstressed conditions: tomato cell contains a pool of free nuclear and cytoplasmic chaperones (shown for HSP70 and HSP90). Heat stress: high temperatures lead to the development of protein aggregates in nucleus and cytoplasm from recruited free chaperones, which causes the loss of their activities. Heat stress and TYLCV infection: massive protein aggregation is observed in nucleus and cytoplasm. Large nuclear aggregates contain HSP70, HSP90 and viral CP together with virions [35,36]. Cytoplasmic aggregates contain CP, but not HSP70 and HSP90.

**Figure 3 plants-11-02944-f003:**
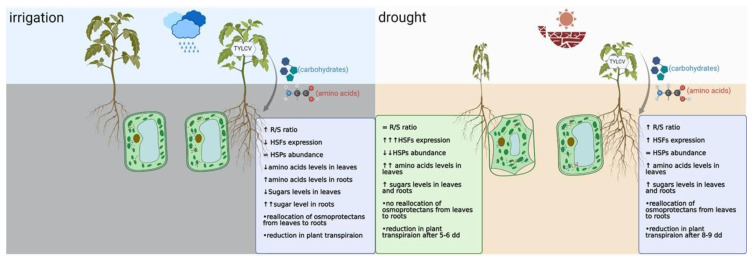
Summary of the main pathways of TYLCV-dependent response of tomatoes to drought. TYLCV infection leads to reallocation of various osmoprotectants, including carbohydrates and amino acids from leaf to root tissues, “preparing” plants to resist drought by developing homeostasis in roots. During growth under water withholding, TYLCV mitigates excessive increase in protectants and, consequently, acute deleterious plant responses [39,41].

**Figure 4 plants-11-02944-f004:**
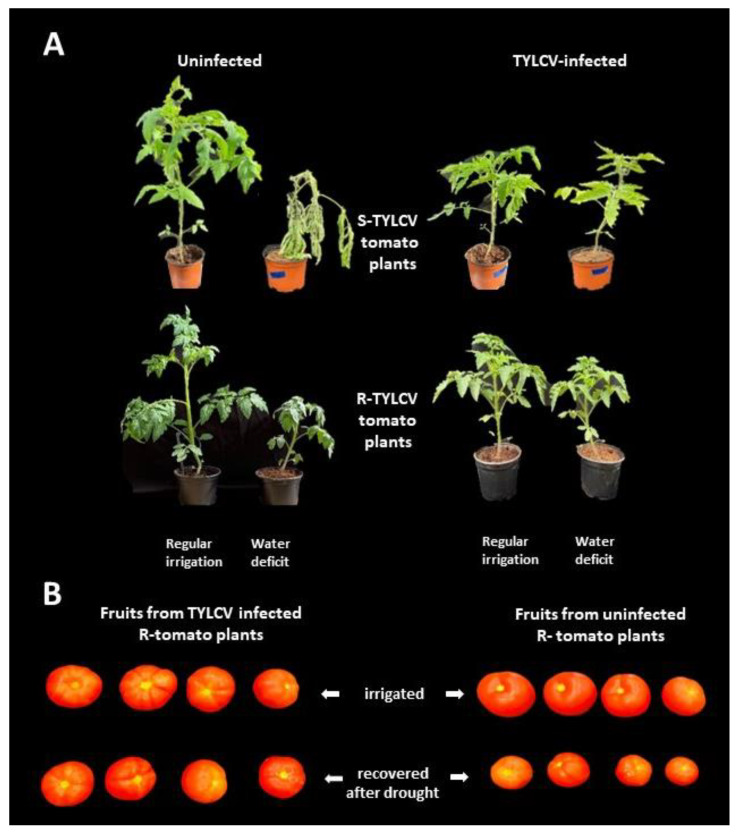
Growth of TYLCV-infected tomatoes under water deficit conditions. Uninfected and TYLCV-infected seedlings of S-TYLCV and R-TYLCV tomatoes were grown under regular irrigation vs. water withholding during 14 days. (**A**). Collapsing of uninfected tomatoes under drought stress, while infected plants survived. (**B**). The yield of infected R-TYLCV tomatoes, grown in water deficit conditions and recovered after normal watering is similar to that of unstressed R-TYLCV plants. The shape and size of individual fruit of experimental tomatoes were comparable. Adapted from [39,41].

## Data Availability

The data presented in this study are available upon request from the corresponding author.

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
