# Peer review of "Exploiting Virus Infection to Protect Plants from Abiotic Stresses: Tomato Protection by a Begomovirus"

_plants, 2022, doi:10.3390/plants11212944_

Round 1

Reviewer 1 Report

This review paper discusses the possibilities of using virus infection as an approach for protection of plants against heat and drought stresses mainly. It is important to discuss this topic as plants are often stressed by multiple stresses at the same time and a combination of different stresses elicits a different response from the individual stresses in isolation.  The discussion is mostly on tomato plants and is based on their own line of work which is mostly on viral infections caused by tomato yellow leaf curl virus (TYLCV) family.

The discussion makes an interesting hypothesis of using virus infection as an approach in controlling heat and drought responses. The authors have related virus infection with protein aggregation. The gene expression changes caused by virus infection are well stated. However, I cannot say the same for gene expression changes which take place against heat and drought stresses. At no point, authors discuss what is the scenario of the complexity of Hsp90 and Hsp70 gene families in tomato. They do not mention anything on gene redundancy in Hsps profiling.  Authors discuss that TYLCV CP protein binds to HsfA2 under the heading number 3 where they have discussed TYLCV and heat stresses. I wish to understand the basis of these protein interactions? Is it a specific interaction? How has this interaction evolved? Hsfs B class members are often considered as negative repressors as compared to HsfA class members which are activators of the heat stress response in Arabidopsis. However, authors have discussed the HsfA and HsfB class members in the same context in their manuscript. Hsp101 is another downstream protein regulated by HsfA class members. However, there is no mention of Hsp101. Hsp101 accumulation has been strongly related to acquisition of heat tolerance. In Section 4 where the authors have discussed TYLCV infection in relation to drought, they have argued that infected plants have reallocation of metabolites from shoot to roots. The authors however do not discuss how exactly this reallocation is regulated by TYLCV?

TYLCV infections can cause huge damage to tomato yield. The argument put forth by the authors that TYLCV infection will protect plants from heat and drought is fine but what about damage inflected by the infection?

However, the arguments put forth by the authors in interactions of viral infection and response to abiotic stresses are interesting and original. The manuscript is well-written. It brings out some areas of work which should be further worked upon, and I recommend this paper for publication.      

Author Response

Reviewer 1

….However, I cannot say the same for gene expression changes which take place against heat and drought stresses. At no point, authors discuss what is the scenario of the complexity of Hsp90 and Hsp70 gene families in tomato. They do not mention anything on gene redundancy in Hsps profiling.

  • Genome-wide expression studies (in vivo and in silico) have shown that heat shock factors (including heat shock proteins and their transcription factors) are encoded by large gene families (about 21 in tomato), which play a role in heat stress responses (Yang et al., PeerJ, 2016; Jacob et al., Plant Biotech J, 2017). Transcriptome sequencing indicated that subjecting tomato cv Moneymaker to short periods (hours) of heat stress was accompanied by an increase of HSP expression, e.g. HSP20 and HSP70 (Su et al., J Plant Biochem Biotech, 2022). However, in the latter study, the redundancies of the HSP genes was not considered. It was neither in earlier HSP-HSFs interaction studies (Hahn et al., Plant Cell, 2011). Added un text.
  • In the case of geminivirus-dependent (TYLCV) regulation of HSPs patterns, the redundancy of HSPs is not a critical topic. TYLCV prevents dramatic changes in hsps expression in plant cells, stabilizing the amount of HSPs. In our earlier studies, HSP stabilization was shown for several selected HSPs (HSP60, 70, 90), especially in R-tomatoes, infected by TYLCV (Gorovits et al., Mol Plant Mic Inter, 2007; Gorovits and Czosnek, Tomato yellow leaf curl virus, 2007). Added in text.

Authors discuss that TYLCV CP protein binds to HsfA2 under the heading number 3 where they have discussed TYLCV and heat stresses. I wish to understand the basis of these protein interactions? Is it a specific interaction? How has this interaction evolved?

  • We have previously described, for the first time, the interaction between HSFA2 and TYLCV CP in tomato (Anfoka et al., Sci Rep, 2016). The His-tagged TYLCV major viral proteins CP, V2, C1, C2, C3, and C4, were over-expressed in coli and bound to His-Bind Resin in Ni-column. Total protein extracts from heat shock-treated (1 h) infected tomatoes (14 dpi) were passed through Ni-columns, each bound to a different His-tagged viral protein. Following elution, the bound proteins were immuno-detected with anti-HSFA2 antibodies. Complexes between HSFA2 and all the six TYLCV proteins were detected. E. coli extracts, passed through the same resins, were used as negative controls to confirm the absence of any background originating from the protein purification procedure. By comparison, HSP70 interacted with only three viral proteins, including CP. To identify HSFA2-TYLCV complexes in the tomato leaf cellular compartments, proteins from separated cytoplasmic and nuclear fractions were passed through the Ni resin columns bound to the viral proteins. Pull down of plant HSFA2-CP complexes were found in nuclear, but not in cytoplasmic, protein extracts. HSFA2-CP binding was not found in E. coli protein extracts. These results indicated that TYLCV CP was able to bind cellular HSFA2, restricting the capacity of free transcription factor to promote heat shock gene transcription in plant nuclei.
  • The presence of HSFA2-CP complexes was confirmed by in situ immuno-fluorescent visualization of plant HSFA2 and viral CP in leaf sections of symptomatic tomato plants at 21 dpi, heat shock treated in vitro for 1 h.
  • The interaction between HSFA2 and TYLCV CP was described in a previous paper of ours (Gorovits et al., Phytopath Res, 2019). Therefore, in the current manuscript, we have omitted the detailed description of the procedures.

Added to text.

Hsfs B class members are often considered as negative repressors as compared to HsfA class members which are activators of the heat stress response in Arabidopsis. However, authors have discussed the HsfA and HsfB class members in the same context in their manuscript.

  • As in the case of HSPs patterns, the down or up-regulation of Hsfs expressions in response to abiotic stresses is attenuated by TYLCV infection to prevent extreme changes, which could lead to cell death.

Hsp101 is another downstream protein regulated by HsfA class members. However, there is no mention of Hsp101. Hsp101 accumulation has been strongly related to acquisition of heat tolerance.

  • HSP101/ClpB chaperone is induced by heat and other stresses in different plants (reviewed by Mishra and Gover, Crit Rev Biotech, 2016). We used anti-HSP101 antibodies (a gift of A. Grover) in our previous study of tomato response to heat shock (Anfoka et al., Sci Rep, 2016). We demonstrated the induction of HSP101 after heat shock; the activation of HSP101 was similar to that of HSFA2 and HSP90 and was dependent on TYLCV amounts contained in tomato tissues. In the current manuscript, the role of HSP70 and HSP90 was discussed not only in response to heat stress, but mainly in the development of protein aggregates. The involvement of HSP101 aggregates in tomatoes infected by TYLCV was not studied, therefore it was not mentioned in the text.

Added to text

In Section 4 where the authors have discussed TYLCV infection in relation to drought, they have argued that infected plants have reallocation of metabolites from shoot to roots. The authors however do not discuss how exactly this reallocation is regulated by TYLCV?

  • The data about metabolite reallocation in TYLCV infected plants was discovered during last two years only, so we did not have enough time to investigate the nature of its regulation. This will be the object of future research.

TYLCV infections can cause huge damage to tomato yield. The argument put forth by the authors that TYLCV infection will protect plants from heat and drought is fine but what about damage inflected by the infection?

  • The use of TYLCV resistant tomato lines, which were developed in cooperation of Israel, Jordan, Egypt, Morocco laboratories solves the problem of potential damage, caused by virus infection. As in was described in section 6, R tomatoes contained TYLCV, but did not show symptoms and produced yields.

Discussed in the text.

Reviewer 2 Report

In this review Gorovotis et al. describe how virus infection can be exploited to alleviate different abiotic stresses in plants. The authors focus on the tomato yellow leaf curl virus-tomato interactions to exemplify different virus-induced tolerances to stressors. The authors have studied this pathosystem previously and they are well-versed in the related literature. The manuscript characterizes a broad set of cases were the virus infection induces tolerance to abiotic stressors. This will be of interest for the researchers studying plant virus-interactions and plant tolerance to stress.

The review will be an interesting contribution to the scientific community but before being published some aspects should be revised:

1) The title is quite general for the specific host-virus interaction that is being reviewed. I suggest something like “Exploiting Begomovirus Infection to Protect Tomatoes from Abiotic Stresses: Tomato Protection by a Begomovirus” or “Exploiting Virus Infection to Protect Plants from Abiotic Stresses: Tomato Protection by a Begomovirus”

2) The structure of the introduction confuses me. It starts with introduction on TYLCV, next paragraph talks about general effects of other viruses and in the next one comes back again to TYLCV. I would start from a broader view and then focus on TYLCV.

3) In the sentence “while cucumber mosaic virus (CMV) changed plant resistance not only to drought, but also to cold” I think the correct verb is to induce and not change.

4) In the citation number 7 the authors cite Gonzalez et al 2021 (PNAS) for indicating that “Viruses were defined as mutual interactors with plant cellular mechanisms, resulting in plant protection against some abiotic stresses”. The correct citation would be Gonzalez et al. 2020 (Advances in Virus Research). Gonzalez et al 2021 just describes how a virus may evolve towards a mutualistic relationship with its plant host and increase its tolerance to drought.

5) The article focusses on TYLCV. Is there any other virus that infects tomatoes that has a similar effect? It would be interesting to mention and briefly discuss if the protective effect happens (or could happen) with other tomato viruses and if the resistant tomato strategy could be applied to other viruses. 

6) In the paragraph “TYLCV does not induce a hypersensitive response and cell death upon whitefly-me-diated infection of S-TYLCV tomato plants, until diseased tomatoes become senescent [22]. Oppositely, the TYLCV capacity to promote enhanced survival of host tomatoes ex-posed to different stresses, has been gradually uncovered. It is summarized in the current review.”

The commas in the underlined text are unnecessary.

7) All the sections are clearly structured around abiotic stressors but “2. Cell death, induced by inactivation of HSP90, is suppressed in TYLCV-infected plants”. I suggest to add an introductory sentence with some citations about known abiotic factors that induce cell death. This is well done in next section (i.e. “Numerous stresses cause protein damage and induce cellular protective mechanisms in order to maintain protein homeostasis.”) but it misses references.

8) About figure 3. I will make more visual the distinction between conditions. Maybe using separate backgrounds and adding some text. 

What plants are those? R-TYLCV ones right?

I am guessing that there are important things going on in the cells but I cannot see them. I would make them bigger, adding bigger brown dots and making it clear that they represent osmoprotectans.

9) About figure 4

(i) Where do the images come from? Does the figure 4 comes from Shteinberg et al. (Cells ,2021) and/or Mishra et al. (Molecular Plant Pathology, 2021)? In that case, mention the figure derives from there.

(ii)In 4B it is necessary to add pictures of fruits after drought to visualize the level of recovery thanks to TYLCV

(iii) In addition, some minor changes to clarify the text in the figure

- Not-infected -> “Not infected” , “Non-infected”, or “Uninfected”

- S-TYLCV -> S-TYCLV tomato plants

- R-TYLCV -> R-TYCLV tomato plants

- Irrigation -> Regular irrigated

- TYLCV-infected R-TYLCV -> TYLCV-infected fruits from R-TYLCV tomato plants

10) “Use of TYLCV abilities to protect tomato crops against environmental stresses”. Viruses do not have abilities. What we humans could do is to use virus infection to induce in the host which makes it tolerate stresses.

11) Please revise the text to improve the flow and clarity of the ideas that want to be transmitted. I feel that sometimes the brackets are over used. See for example this paragraph:

All modern agricultural plants are the result of selection and improvement of wild relatives of the cultivated species. While the wild relatives have acquired tolerance to the major abiotic and biotic stresses, adapting themselves to harsh environments, they are not edible (sometimes poisonous). During the Neolithic agricultural revolution (about 10,000 years ago) selection for better, tastier, edible plants (not yield) has begun. The advent of modern breeding in the last 300 years saw the selection for better fruits and higher yields (beautiful and plenty) to feed increasing populations after the black death and other epi-demics in Europe, Asia and the Americas.”

I would suggest something like this:

“All modern agricultural plants are the result of selection and improvement of wild relatives of the cultivated species. While the wild relatives have acquired tolerance to the major abiotic and biotic stresses, the adaption to harsh environments made them not edible and sometimes even poisonous. During the Neolithic agricultural revolution (about 10,000 years ago) the selection for better, tastier, edible plants began. The selection for yield did not start until the advent of modern breeding in the last 300 years saw, when the selection for more visual-appealing fruits and higher yields to feed increasing populations started after the black death and other epidemics in Europe, Asia and the Americas.”

12) In the conclusion sentence “If man disappears for any reason (cli-mate change, war, migration), the plants will die and the region will become a desert.”

First, I would specify the is not the plants but the crops. Second, not all crops will disappear without human intervention. Third, if the crops disappear the consequence will be that the crops will disappear and other plant species would take over. The area where the crops disappear does not necessarily need to become a desert, right? Finally, I understand the point but I (personally) find the “If man disappears for any reason (cli-mate change, war, migration)” quite a bit dramatic. It may be better just to mention a general lack of human intervention will make some agricultural species disappear.

13) In “Genetic engineering: by influencing the metabolic pathways to increase resilience.” There are other mechanisms apart from metabolic ones that could be engineered.

14) In “Whether fruit production of pre-inoculation of R-TYLCV tomato seedlings in field and greenhouses”. About the approach on the field I think there should be at least some mention about the possibilities of outbreaks from a resistant plant to sensitive genotypes (or other species). Especially if we consider that TYLCV is a vector transmitted virus.

Author Response

Reviewer 2

The review will be an interesting contribution to the scientific community but before being published some aspects should be revised:

The title is quite general for the specific host-virus interaction that is being reviewed. I suggest something like “Exploiting Begomovirus Infection to Protect Tomatoes from Abiotic Stresses: Tomato Protection by a Begomovirus” or “Exploiting Virus Infection to Protect Plants from Abiotic Stresses: Tomato Protection by a Begomovirus”.

- The title has been changed (thank you for the suggestion) to “Exploiting Virus Infection to Protect Plants from Abiotic Stresses: Tomato Protection by a Begomovirus”

The structure of the introduction confuses me. It starts with introduction on TYLCV, next paragraph talks about general effects of other viruses and in the next one comes back again to TYLCV. I would start from a broader view and then focus on TYLCV.

- The order has been changed according to the recommendation of the Reviewer.

In the sentence “while cucumber mosaic virus (CMV) changed plant resistance not only to drought, but also to cold” I think the correct verb is to induce and not change.

- Corrected.

In the citation number 7 the authors cite Gonzalez et al 2021 (PNAS) for indicating that “Viruses were defined as mutual interactors with plant cellular mechanisms, resulting in plant protection against some abiotic stresses”. The correct citation would be Gonzalez et al. 2020 (Advances in Virus Research). Gonzalez et al 2021 just describes how a virus may evolve towards a mutualistic relationship with its plant host and increase its tolerance to drought.

 - The reference has been changed to Gonzalez et al., 2020.

The article focusses on TYLCV. Is there any other virus that infects tomatoes that has a similar effect?

  • Not to our knowledge.

It would be interesting to mention and briefly discuss if the protective effect happens (or could happen) with other tomato viruses and if the resistant tomato strategy could be applied to other viruses.

  • Virus mediated stress tolerance is a broad phenomenon described for several ssRNA viruses belonging to different taxonomic families and infecting a variety of susceptible host plants (recently reviewed by Aguilar and Lozano-Duran, Stress Biol, 2022). TYLCV was cited as the only begomovirus tested so-far. Begomoviruses are so deleterious that using susceptible plants will not provide the desired protection effect because of virus infection-caused damages. Using tomato begomoviruses others than those belonging to the TYLCV family is an attractive working hypothesis. Several begomoviruses infect cultivated tomatoes: tomato mottle leaf curl virus, tomato leaf curl virus, tomato golden mosaic virus. To prove the concept, tomato cultivars will have to be screened for tolerance to these viruses to identify the suitable genotypes. This strategy could be applied to other viruses, preferably with crops, provided that the varieties used are tolerant to the virus, not to damage the plant before a putative effect could be observed.

In the paragraph “TYLCV does not induce a hypersensitive response and cell death upon whitefly-mediated infection of S-TYLCV tomato plants, until diseased tomatoes become senescent [22]. Oppositely, the TYLCV capacity to promote enhanced survival of host tomatoes ex-posed to different stresses has been gradually uncovered. It is summarized in the current review.” The commas in the underlined text are unnecessary.

- Corrected.

All the sections are clearly structured around abiotic stressors but “2. Cell death, induced by inactivation of HSP90, is suppressed in TYLCV-infected plants”. I suggest to add an introductory sentence with some citations about known abiotic factors that induce cell death.

  • Two references are added.

This is well done in next section (i.e. “Numerous stresses cause protein damage and induce cellular protective mechanisms in order to maintain protein homeostasis.”) but it misses references.

  • The sentence “Cell death can often be activated as a defense response against biotic and abiotic stresses” was added to the beginning of section 2. The sentence “Numerous stresses cause protein damage and induce cellular protective mechanisms in order to maintain protein homeostasis” is to global to start to add specific references, which are numerous. While answering to this Reviewer, we bounced into a very recent review that provides a glimpse into the physiological responses to abiotic stresses (dos Santos et al., Stresses, 2022).

About figure 3. I will make more visual the distinction between conditions. Maybe using separate backgrounds and adding some text. I am guessing that there are important things going on in the cells but I cannot see them. I would make them bigger, adding bigger brown dots and making it clear that they represent osmoprotectants.

  • The colors have been changed, the sizes of root cells enlarged.

What plants are those? R-TYLCV ones right?

  • Yes, it is R-TYLCV tomatoes.

About figure 4, where do the images come from? Does the figure 4 comes from Shteinberg et al. (Cells ,2021) and/or Mishra et al. (Molecular Plant Pathology, 2021)? In that case, mention the figure derives from there.

  • The data used in the figures has been derived from two recent publications of ours Shteinberg et al. 2021 and Mishra et al. 2021, as mentioned by the Reviewer.

In 4B it is necessary to add pictures of fruits after drought to visualize the level of recovery thanks to TYLCV.

  • This has been modified accordingly.

In addition, some minor changes to clarify the text in the figure

  • Not-infected -> “Not infected” , “Non-infected”, or “Uninfected” --- done
  • S-TYLCV -> S-TYCLV tomato plants --- done
  • R-TYLCV -> R-TYCLV tomato plants --- done
  • Irrigation -> Regular irrigated --- done
  • TYLCV-infected R-TYLCV -> TYLCV-infected fruits from R-TYLCV tomato plants --- done

Use of TYLCV abilities to protect tomato crops against environmental stresses”. Viruses do not have abilities. What we humans could do is to use virus infection to induce in the host which makes it tolerate stresses.

  • Corrected accordingly.

Please revise the text to improve the flow and clarity of the ideas that want to be transmitted. I feel that sometimes the brackets are over used. See for example this paragraph:

“All modern agricultural plants are the result of selection and improvement of wild relatives of the cultivated species. While the wild relatives have acquired tolerance to the major abiotic and biotic stresses, adapting themselves to harsh environments, they are not edible (sometimes poisonous). During the Neolithic agricultural revolution (about 10,000 years ago) selection for better, tastier, edible plants (not yield) has begun. The advent of modern breeding in the last 300 years saw the selection for better fruits and higher yields (beautiful and plenty) to feed increasing populations after the black death and other epidemics in Europe, Asia and the Americas.”

I would suggest something like this:

“All modern agricultural plants are the result of selection and improvement of wild relatives of the cultivated species. While the wild relatives have acquired tolerance to the major abiotic and biotic stresses, the adaption to harsh environments made them not edible and sometimes even poisonous. During the Neolithic agricultural revolution (about 10,000 years ago) the selection for better, tastier, edible plants began. The selection for yield did not start until the advent of modern breeding in the last 300 years saw, when the selection for more visual-appealing fruits and higher yields to feed increasing populations started after the black death and other epidemics in Europe, Asia and the Americas.”

  • Has been changed according to the suggestion of the Reviewer.

In the conclusion sentence“If man disappears for any reason (climate change, war, migration), the plants will die and the region will become a desert.”

First, I would specify the is not the plants but the crops. Second, not all crops will disappear without human intervention. Third, if the crops disappear the consequence will be that the crops will disappear and other plant species would take over. The area where the crops disappear does not necessarily need to become a desert, right? Finally, I understand the point but I (personally) find the “If man disappears for any reason (cli-mate change, war, migration)” quite a bit dramatic. It may be better just to mention a general lack of human intervention will make some agricultural species disappear. ---

  • We agree with the Reviewer, that this paragraph is overly dramatic. We just wished to mention that the extreme attention of man is needed because most domesticated crops are very susceptible to environmental stresses. Once this attention is discontinued for any reason (e.g. environmental changes, wars), the crops are endangered and may well disappear. Several historical examples corroborate this view, for example Mayan and Akkadian civilizations.

In “Genetic engineering: by influencing the metabolic pathways to increase resilience.” There are other mechanisms apart from metabolic ones that could be engineered.

  • Aguilar and Lozano-Duran (cited above) suggest to overexpress in transgenic plants those viral proteins that have been shown to be involved in tolerance to abiotic stresses, such as 2b in CMV, C4 in TYLCV, or P25 in potato virus X.

In “Whether fruit production of pre-inoculation of R-TYLCV tomato seedlings in field and greenhouses”. About the approach on the field I think there should be at least some mention about the possibilities of outbreaks from a resistant plant to sensitive genotypes (or other species). Especially if we consider that TYLCV is a vector transmitted virus.

  • We have done that and cited possible spread of TYLCV from tomato to pepper (Lobin et al, 2022). It might be possible to pre-inoculating (by agroinfection) seedlings with a TYLCV symptomless mutant lacking 20 amino acids near the N-terminus of the CP and therefore not transmissible by whiteflies (Peretz et al., Plant Physiol, 2007) before planting in the field.

Discussed in the text.

Reviewer 3 Report

The review article by Gorovits et al, highlights the potential use of virus infection to make plants tolerant to abiotic stress which is currently a highly active research field. It is important to the journal readers however, the article fails to develop the importance of virus infection that readers can appreciate from different research fields. 

Major critical comments:

1- The article did a good review of the articles but failed to provide critical research gaps and future research direction which requires citing more relevant articles.

2- Organic farming and efficient water usage in agriculture need to be discussed with proper citation of new articles. Recently, there were articles published in a special issue of Environmental Science and Pollution Research journal.

3- All the figures can be improved significantly. Currently, it is very confusing and not all the symbols are described in the legend. In figure 1, it is better to show in each scenario how protein ubiquitination level and degradation vary to make the figures relatable to the text and easier to understand. 

4- in fig 1 and 2, HSP90 is indicated by different colours. Please be consistent with the figures. In figure 2, heat stress + TYLCV infection, the two aggregates found in the nucleus and cytoplasm are different. So, please use different symbols or colours to represent these aggregates. 

5- Though the R-TYLCV plants don't show any symptoms, it should be checked what is the population diversity of the virus as it may lead to virus mutants that will be highly virulent in other crops through spillover. The work done at Fernando Garcia-Arenal lab on virus evolution should be discussed here where they showed tobamovirus isolates from a wild plant, N. glauca is more diverse than in pepper plants. In wild plants virus don't show any symptoms and highly virulent in pepper. These interactions might happen in R-TYLCV plants and should be investigated in the future.

Author Response

Reviewer 3

The article did a good review of the articles but failed to provide critical research gaps and future research direction which requires citing more relevant articles.

  • We have done this. See also reviewer 1.

2- Organic farming and efficient water usage in agriculture need to be discussed with proper citation of new articles. Recently, there were articles published in a special issue of Environmental Science and Pollution Research journal.

  • We have discussed water usage, water scarcity and water recycling, and added several references. Parizad, S.; Bera, S. The effect of organic farming on water reusability, sustainable ecosystem, and food toxicity. Environ Sci Pollut Res. 2021. https://doi.org/10.1007/s11356-021-15258-7. Christou, A., Karaolia, P., Hapeshi, E., Michael, C.; Fatta-Kassinos, D. Long-term wastewater irrigation of vegetables in real agricultural systems: concentration of pharmaceuticals in soil, uptake and bioaccumulation in tomato fruits and human health risk assessment. Water Res. 2017, 109, 24–34.

3- All the figures can be improved significantly. Currently, it is very confusing and not all the symbols are described in the legend. In figure 1, it is better to show in each scenario how protein ubiquitination level and degradation vary to make the figures relatable to the text and easier to understand.

  • The figures have been changed accordingly.

4- in fig 1 and 2, HSP90 is indicated by different colours. Please be consistent with the figures. In figure 2, heat stress + TYLCV infection, the two aggregates found in the nucleus and cytoplasm are different. So, please use different symbols or colours to represent these aggregates.

  • The figures have been changed accordingly.

5- Though the R-TYLCV plants don't show any symptoms, it should be checked what is the population diversity of the virus as it may lead to virus mutants that will be highly virulent in other crops through spillover. The work done at Fernando Garcia-Arenal lab on virus evolution should be discussed here where they showed tobamovirus isolates from a wild plant, N. glauca is more diverse than in pepper plants. In wild plants virus don't show any symptoms and highly virulent in pepper. These interactions might happen in R-TYLCV plants and should be investigated in the future.

  • Virus evolution is indeed a fascinating topic. Viruses can adapt to a new host by modifying its genome to favor, for example, improved replication. It can also modify itself under various environmental stresses. Ecological selection of new virus variants has been recently discussed; they usually are not the result of the response of the host to infection but rather of changes affecting the spread of the virus insect vector or changes in vector populations following drastic environmental changes (McLeish et al., 2021). McLeish, M.; Fraile, A.; Garcia-Arenal, F. Population genomics of plant viruses: the ecology and evolution of virus emergence. Phytopathology, 2021, 111, 32-39. The mutation rate of TYLCVs in its tomato host has been reported to be in the order of 10-4 substitutions per site during the 45 days of the experiment. Most relevant to our work, the rate of mutation in TYLCV-susceptible and resistant genotypes was not significantly different (Sánchez-Campos, S.; Domínguez-Huerta, G.; Díaz-Martínez, L.; Tomás, D.M.; Navas-Castillo, J.; Moriones, E.; Grande-Pérez, A. Differential shape of geminivirus mutant spectra across cultivated and wild hosts with invariant viral consensus sequences. Front Plant Sci, 2018, 9, 932).

Discussed in the text.

Round 2

Reviewer 2 Report

I thank the authors for the effort they put revising the manuscript. They have addressed all my comments and I find the resulting article much improved and ready for acceptance.

Reviewer 3 Report

The manuscript looks good and is ready to publish without any reservations.